# LaRa: Latents and Rays for Multi-Camera Bird's-Eye-View Semantic Segmentation

**Florent Bartoccioni**
Valeo.ai    Inria*

**Éloi Zablocki**
Valeo.ai

**Andrei Bursuc**
Valeo.ai

**Patrick Pérez**
Valeo.ai

**Matthieu Cord**
Valeo.ai
Sorbonne Université

**Karteek Alahari**
Inria*

**Abstract:** Recent works in autonomous driving have widely adopted the bird's-eye-view (BEV) semantic map as an intermediate representation of the world. Online prediction of these BEV maps involves non-trivial operations such as multi-camera data extraction as well as fusion and projection into a common top-view grid. This is usually done with error-prone geometric operations (e.g., homography or back-projection from monocular depth estimation) or expensive direct dense mapping between image pixels and pixels in BEV (e.g., with MLP or attention). In this work, we present 'LaRa', an efficient encoder-decoder, transformer-based model for vehicle semantic segmentation from multiple cameras. Our approach uses a system of cross-attention to aggregate information over multiple sensors into a compact, yet rich, collection of latent representations. These latent representations, after being processed by a series of self-attention blocks, are then reprojected with a second cross-attention in the BEV space. We demonstrate that our model outperforms the best previous works using transformers on nuScenes. The code and trained models are available at https://github.com/valeoai/LaRa.

**Keywords:** bird's eye view semantic segmentation; encoder-decoder transformers

## 1 Introduction

To plan and drive safely, autonomous cars need accurate 360-degree perception and understanding of their surroundings from multiple and diverse sensors, e.g., cameras, RADARs, and LiDARs. Most of the established approaches tardily aggregate independent predictions from each sensor [1, 2, 3]. Such a late fusion strategy has limitations for reasoning globally at the scene level and does not take advantage of the available prior geometric knowledge that links sensors. Alternatively, the bird's-eye-view's (BEV) representational space, a.k.a. top-view occupancy grid, recently gained considerable interest within the community. BEV appears as a suitable and natural space to fuse multiple views [4, 5] or sensor modalities [6, 7] and to capture semantic, geometric, and dynamic information. Besides, it is a widely adopted choice for downstream driving tasks including motion forecasting [5, 8, 9, 10] and planning [11, 12, 13, 14]. In this paper, we focus on BEV perception from multiple cameras. The online estimation of BEV representations is usually done by: (i) imposing strong geometric priors such as a flat world [15] or correspondence between pixel columns and BEV rays [16], (ii) predicting depth probability distribution over pixels to lift from 2D to 3D and project back in BEV [4, 5], a system subject to compounding errors, or, (iii) learning a costly dense mapping between multi-camera features and the BEV grid pixels [17].

Here, we depart from these dominant strategies and introduce 'LaRa', a novel transformer-based model for vehicle segmentation from multiple cameras. In contrast to prior works, we propose to use a latent 'internal representation' instantiated as a collection of vectors. Fusing multiple views into a

---

*Univ. Grenoble Alpes, Inria, CNRS, Grenoble INP, LJK, 38000 Grenoble, France

6th Conference on Robot Learning (CoRL 2022), Auckland, New Zealand.

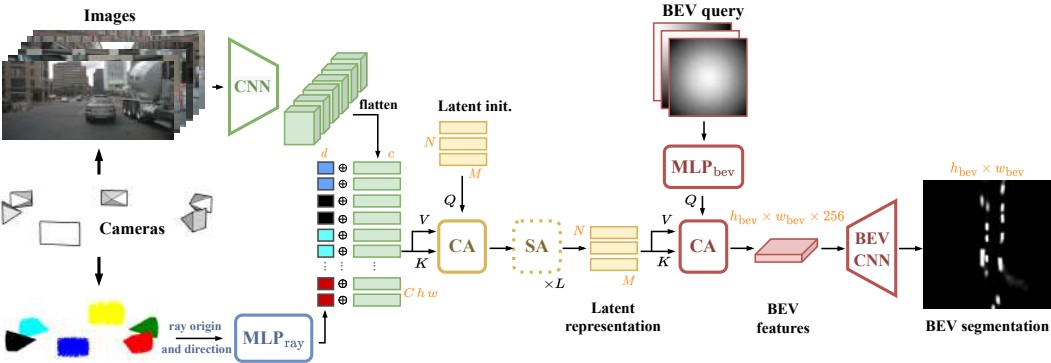

Figure 1: **LaRa overview.** Semantic features (green) are extracted from the images with a shared CNN and are concatenated with ray embeddings (multi-colored) that inform about geometric information to spatially relate pixels within and across cameras. This representation is then fused into a compact latent representation through one cross-attention (CA) and $L$ self-attention (SA) layers (yellow). The final BEV map is obtained by querying the latent representation with a cross-attention and then refined with BEV CNN (red). $\oplus$ denotes concatenation. The orange letters indicate tensor dimensions. $K$, $Q$, and $V$ are the *Key*, *Query*, and *Value* of the cross-attentions.

compact latent space comes with several benefits. First, it provides an explicit control on the memory and computation footprint of the model, instead of the quadratic scaling of the full mapping between multi-camera features and the BEV grid pixels [17]. By design, the number of latents that we use is much smaller compared to the spatial resolution of the BEV grid, enabling a highly-efficient aggregation of information at the latent-level while exploiting spatial cues within and across camera views. Moreover, we also hypothesize that discarding error-prone modules in the pipeline such as depth estimation [4, 5] can boost model accuracy and robustness. Finally, we can directly predict at the full-scale BEV resolution bypassing noisy upsampling operations. This is infeasible, within a reasonable computational budget, for prior works restricted to coarser BEV grids as they map densely between all the image and BEV grid pixels [17]. Besides, as an orthogonal contribution, we augment input features with ray embeddings that encode geometric relationships within and across images. We show that such spatial embeddings, encoding prior geometric knowledge, help guide the cross-attention between input features and the latent vectors.

Our approach is extensively validated against prior works on the nuScenes [10] dataset. We significantly improve the performance on the vehicle segmentation task, outperforming recent high-performing models [4, 17]. Moreover, we show interesting properties of our cross-attention, which naturally stitches multiple cameras together. We also perform several ablation and sensitivity studies of our architecture with respect to hyper-parameters changes. Overall, LaRa is a novel model that learns the mapping from camera views to bird's-eye-view for the task of vehicle semantic segmentation. In summary, our contributions are as follows:

- We encode multiple views into a compact latent space that enables precise control on the model's memory and computation footprint, decoupled from the input size and output resolution.

- We augment semantic features with spatial embeddings derived from cameras' calibration parameters and show that it strongly helps the model learn to stitch multiple views together.

- Our architectural contributions are validated on nuScenes where we reach new SOTA results.

## 2   Related work

### 2.1   BEV semantic segmentation

Models for BEV segmentation are typically structured in two parts. They first extract features of each camera and then project them into a common top-view grid, called the bird's-eye-view. There are different strategies for this projection, which can be grouped into the following categories.

**IPM-based.** Inverse perspective mapping (IPM) defines the correspondence between the camera and the ground planes as a homography matrix. IPM makes strong assumptions that the world is planar and the cameras' horizontal axes are parallel to the ground. Early works [18, 19] apply it directly to raw camera pixels or features. This approach suffers from blurring and stretching artifacts for distant objects (as they have fewer pixels in the camera view) and objects with a height (as they violate the planar world assumption). To alleviate these shortcomings, a generative adversarial network [20] or training a BEV decoder with synthetic ground-truth [15] has been used to refine the IPM projection.

**'Lift-splat'-based: guiding with depth.** Using depth information to lift features from 2D to 3D and then 'splatting' them in BEV space recently gained popularity for its effectiveness and sound geometric definition. Among the formulations of depth estimation for BEV projection [2, 4, 5, 21, 22], estimating depth probabilities along camera rays appears to perform the best [4, 5]. However, such a strategy, depth being the most influential factor [23], is subject to compounding errors. Inaccuracies in depth prediction will propagate into the BEV features, which themselves can be erroneous.

**Implicitly learned with dense networks.** An alternative to explicit geometric projection is to learn the mapping from data. For instance, VPN [24] uses an MLP to make a dense correspondence between pixels in the camera views and BEV. These methods rely on such expensive operations and do not use readily available spatial information given by the calibrated camera rig capturing the images. The BEV projection must be entirely learned, and as it is determined by training data, it can hardly apply to new settings with slightly different camera calibrations. Alternatively, PON [16] builds on the observation that a column in the camera image contains all the information of the corresponding ray in BEV: it first encodes each column into a feature vector, which is then decoded into a ray along the depth dimension. However this relies on two implicit assumptions: (i) the camera follows a pinhole projective model, and (ii) it is horizontally aligned with the ground plane.

**Implicitly learned with transformer architectures.** The attention system at the core of transformer architectures [25, 26, 27, 28] allows learning of long-range dependencies and correspondences explicitly. These architectures have recently been employed for the BEV semantic segmentation task, yielding among the best-performing methods [17, 29, 30]. Nonetheless, a direct cross-attention [25] between camera images and the BEV grid is computationally expensive. BEVFormer [29] alleviates this issue by only cross-attending BEV pixels with cameras in which the BEV pixel is visible and by replacing the heavier multi-head attention [25] with deformable attention [31]. CVT [17] keeps the vanilla multi-head cross-attention [25] but applies it between low-resolution camera feature maps and a small BEV grid which is then upsampled to reach the final resolution. GitNet [30] restrains the cross-attentions to column-ray pairs making the same original implicit assumptions as PON [16]. Our proposed model LaRa belongs to this category as it learns the BEV representation with a transformer architecture. On the other hand, our attention scheme does not impose strong geometric assumptions while still being efficient enough to attend to a full-resolution BEV grid.

## 2.2 Incorporating geometric priors in Transformers

Since transformer architectures are permutation-invariant, spatial relationships between image regions are lost if no precautions are taken. A standard practice to retain this spatial knowledge is to add a positional embedding to the input of attention layers [25]. A popular approach is to encode the position of pixels with sine and cosine functions of varying frequencies [25, 27, 28] applied over the horizontal and vertical axes. An alternative solution to induce spatial awareness in the model is to concatenate $x, y$ positions to feature maps fed to convolutional layers [32].

Related to our ray embedding proposition, recent works [33, 17] embed the parameters of the calibrated cameras in the image features, improving training efficiency and segmentation performance. Similar to LaRa, IIB [33] also encodes the camera center and ray direction in the input feature sequence, but it applies it to depth estimation on image pairs in an indoor environment. Furthermore, Yifan et al. [33] embed the origin and direction of rays into Fourier features, which can become memory intensive depending on the number of frequency bands and also introduces additional hyper-parameters to tune. CVT [17] adds up a ray direction embedding to the input feature sequence, but, differently from ours, uses the camera center embedding in the BEV query. This requires a BEV query and 'cross-view attention' operation per camera, increasing the memory and computational footprint, thus limiting the maximum resolution of the BEV query.

# 3 LaRa: Our Latents and Rays Model

Given multiple cameras observing the scene, our goal is to estimate a binary occupancy grid [34] $\hat{y} \in \{0, 1\}^{h_{\text{bev}} \times w_{\text{bev}}}$ of size $h_{\text{bev}} \times w_{\text{bev}} \in \mathbb{N}^2$ for vehicles in the surroundings of the ego car. We propose 'LaRa' a transformer-based architecture to efficiently aggregate information gathered from multiple cameras into a compact latent representation before expanding back into the BEV space. Besides, as we believe that the geometric relationship between cameras should guide the fusion across each camera view, we propose to augment each pixel with the geometry of the ray that captured it. The LaRa architecture is illustrated in Figure 1.

## 3.1 Input modeling with geometric priors

We consider $C$ cameras described by $(I_k, \mathcal{K}_k, \mathcal{R}_k, t_k)_{k=1}^C$, with $I_k \in \mathbb{R}^{H \times W \times 3}$ the image produced by camera $k$, $\mathcal{K}_k \in \mathbb{R}^{3 \times 3}$ the intrinsics, $\mathcal{R}_k \in \mathbb{R}^{3 \times 3}$ and $t_k \in \mathbb{R}^3$ the extrinsic rotation and translation respectively. From these inputs, two complementary types of information are extracted: semantic information from raw images and geometric cues from the camera calibration parameters.

**Semantic information from raw images.** A shared image-encoder $E$ extracts feature maps for each image $F_k = E(I_k) \in \mathbb{R}^{h \times w \times c}$. Following [4, 5], we instantiate $E$ with a pretrained Efficient-Net [35] backbone to produce the multi-camera features. These spatial feature maps in $\mathbb{R}^{C \times h \times w \times c}$ are then rearranged as a sequence of feature vectors, in $\mathbb{R}^{(C\,h\,w) \times c}$.

**Leveraging geometric priors.** To enrich camera features with geometric priors, commonly used sine and cosine spatial embeddings [25, 27, 28] are ambiguous in presence of multiple cameras. A straightforward solution would be to use camera-dependant learnable embeddings in addition to the Fourier embeddings to disambiguate between cameras. However, in our setting, we argue that the geometric relationship between cameras, which is defined by the structure of the camera rig, is crucial to guide the fusion of the views. This motivates our choice to leverage the cameras' extrinsics and intrinsics to encode the position and orientation of each pixel in the vehicle ego-frame.

More precisely, we encode the camera calibration parameters by constructing the viewing ray for each pixel of the cameras. Given a pixel coordinate $x \in \mathbb{R}^2$ within a camera image $I_k$, the direction $d_k(x) \in \mathbb{R}^3$ of the ray that captured $x$ is computed with:

$$d_k(x) = \mathcal{R}_k^{-1} \mathcal{K}_k^{-1} \tilde{x}, \tag{1}$$

where $\tilde{x}$ are the homogeneous coordinates of $x$, and $d_k(x)$ is expressed in ego-coordinates. The origin of the ray $d_k(x)$ is the camera center given by $t_k$.

Then, to fully describe the position and the orientation of the ray that captured pixel $x$, we use the embedding $ray_k(x) \in \mathbb{R}^d$ computed as follows:

$$ray_k(x) = \text{MLP}_{\text{ray}}(t_k \oplus d_k(x)), \tag{2}$$

where $\oplus$ is a concatenation operation and $\text{MLP}_{\text{ray}}$ a 2-layer MLP with GELU activations [36]. The computation is consistent within and across cameras and it exhibits an interesting property: overlapping regions for two cameras with the same optical center have the same ray embedding. Note that the intrinsics are scaled according to the difference in resolution between $I_k$ and $F_k$.

As shown in Figure 1, the final input vector sequence, in $\mathbb{R}^{(C\,h\,w) \times (d+c)}$, is produced by concatenating each of the $C\,h\,w$ feature vectors $F_k(x) \in \mathbb{R}^c$ with its geometric embedding $ray_k(x) \in \mathbb{R}^d$.

## 3.2 Building latent representations and deep fusion

To control the computational and memory footprint of the image-to-BEV block, we leverage findings from general-purpose architectures [28] and propose to use an intermediate fixed-sized latent space instead of learning the quadratic all-to-all correspondence between multi-camera features and BEV space [17]. Formally, the visual representations $F_k$ from all cameras, along with their corresponding geometric embeddings $ray_k$, are compressed by cross-attention [25] into a collection of $N$ learnable latent vectors of dimension $M \in \mathbb{N}$ and processed by a series of $L$ self-attention blocks [25] (see yellow elements in Figure 1). We stress that $N \ll C\,h\,w$, which enables to fuse and process efficiently the visual information coming from all the cameras regardless of the input feature resolution or the number of cameras. Thanks to latent-based querying, this formulation decouples

Table 1: **Intersection-over-Union (IoU) for vehicle segmentation on nuScenes.** 'Setting 1' refers to a 100m×50m grid with a 25cm resolution and 'Setting 2' to a 100m×100m grid with a 50cm resolution. For training and validation, vehicles are considered only if their visibility level is above a predefined threshold (either 0% or 40%). To compare against other works, we refer the reader to Lift-splat [4] and CVT [17].

| Method | Conference | visibility > 0%
Setting 2 | visibility > 40%
Setting 1 | Setting 2 |
|---|---|---|---|---|
| Lift-splat [4] | ECCV'20 | 32.1 | — | — |
| CVT [17] | CVPR'22 | — | 37.5 | 36.0 |
| LaRa (ours) | — | **35.4** | **41.4** | **38.9** |

the network's deep multi-view processing from the input and output resolution. Our architecture can thus take advantage of the full resolution of the BEV grid.

### 3.3 Generating BEV output from latents

The final step is to decode the binary segmentation prediction $\hat{y} \in \{0,1\}^{h_{\text{bev}} \times w_{\text{bev}}}$ from the latent space. In practice, the latent vectors are cross-attended [25] with a BEV 'query' grid $Q \in \mathbb{R}^{h_{\text{bev}} \times w_{\text{bev}} \times d_{\text{bev}}}$ at the final prediction resolution, with $d_{\text{bev}} \in \mathbb{N}$ a hyper-parameter (illustrated by the red blocks in Figure 1). Each element of the query grid is a feature vector encoding the spatial position in the bird's-eye-view which specifies what information the cross-attention would extract from the latent representations. This last cross-attention yields a feature map in BEV space, in dimension $h_{\text{bev}} \times w_{\text{bev}} \times 256$, that is further refined with a small convolutional encoder-decoder U-Net ('BEV CNN' in Figure 1) to finally predict the binary bird's-eye-view semantic map $\hat{y} \in \{0,1\}^{h_{\text{bev}} \times w_{\text{bev}} \times 1}$.

Specifically, we consider a combination of two types of queries: normalized coordinates in the BEV space and radial distance. Normalized coordinates encode ego-centered normalized coordinates of the BEV plane. They are obtained with:

$$Q_{\text{coords}}[i,j] = \left( \frac{2i}{h_{\text{bev}}-1} - 1, \frac{2j}{w_{\text{bev}}-1} - 1 \right), \ \forall i,j \in \{0,\ldots,h_{\text{bev}}-1\} \times \{0,\ldots,w_{\text{bev}}-1\}. \quad (3)$$

Normalized radial distances are simply Euclidean distances of pixels w.r.t. the origin:

$$Q_{\text{radial}}[i,j] = \sqrt{Q_{\text{coords}}[i,j]_i^2 + Q_{\text{coords}}[i,j]_j^2}. \quad (4)$$

While the network could produce a similar embedding from $Q_{\text{coords}}$ using $\text{MLP}_{\text{bev}}$, we find that introducing these radial embeddings along $Q_{\text{coords}}$ empirically improves results. Moreover, this query decoding choice compares favorably against more classical Fourier embeddings [25, 28, 33] and learned query embeddings [25, 27], as shown in Table 2.

## 4 Experiments

**Dataset.** We conduct experiments on the nuScenes dataset [10], which contains 34k annotated sets of frames captured by $C{=}6$ synchronized cameras covering the 360° field of view around the ego vehicle. The extrinsics and intrinsics calibration parameters are given for all cameras in every scene. Raw annotations come in the form of 3D bounding boxes that are simply rendered in the discretized top-down view of the scenes to form the ground-truth for our binary semantic segmentation task.

**Precise settings for training and validation.** With no established benchmarks to precisely compare model's performances, there are almost as many settings as there are previous works. Differences are found at three different levels: The *resolution* of the output grid, the *level of visibility* used to select objects as part of the ground-truth, and the task considered. In this paper, we address the task of *binary semantic segmentation* of all vehicles (`cars`, `bicycles`, `trucks`, *etc.*) [4, 17]. This choice is made to have fair and consistent comparisons with our baselines [4, 17], however, it should be noted that our model is not constrained to this setting. To enable and ease future comparison, we

Table 2: **Ablation study for the input and output query embedding**. Training and evaluation are done in Setting 2 (100m×100m at 50cm resolution), with a visibility $> 0\%$.

| Input geometry embedding | | | | Output query embedding | | | | |
|---|---|---|---|---|---|---|---|---|
| Cam. rays | Cam. idx | Fourier | **IoU** | Radial dist. | Norm. coords | Fourier | Learned | **IoU** |
| ✔ | ✗ | ✗ | **35.4** | ✔ | ✔ | ✗ | ✗ | **35.4** |
| ✔ | ✔ | ✔ | 34.4 | ✗ | ✔ | ✗ | ✗ | 35.1 |
| ✗ | ✔ | ✔ | 32.3 | ✗ | ✗ | ✔ | ✗ | 30.6 |
| ✗ | ✗ | ✔ | 30.5 | ✗ | ✗ | ✗ | ✔ | 21.8 |

have published our code[2]. We also present additional settings in the supplementary material. In all the settings we considered, models are evaluated with the IoU metric.

**Training and implementation details.** We train our model by optimizing the Binary Cross Entropy with our predicted soft segmentation maps and the binary ground-truth. Images are processed at resolution $224 \times 480$. We use the AdamW [37] optimizer with a constant learning rate of $5e{-}4$ and a weight decay of $1e{-}7$. We train our model on 4 Tesla V100 16GB GPUs with a total batch size of 8 for 30 epochs. Training takes on average 11 hours. We use an EfficientNet-B4 [35] with an output stride of 8 as our CNN image encoder. For the BEV CNN we follow Philion and Fidler [4]. $\text{MLP}_{\text{bev}}$ is a 2-layer MLP producing $d_{\text{bev}} = 128$-dimensional features.

## 4.1 Comparison with previous works

In Table 1, we compare the IoU performances of LaRa against two baselines Lift-Splat [4] and CVT [17] on vehicle BEV segmentation in their respective training and evaluation setups. In all cases, we improve results by a significant margin. More precisely, we improve by 10% compared to Lift-Splat in their settings, by 10% and 8% compared to CVT respectively in Setting 1 and Setting 2. This suggests that our model can better extract the geometric and semantic information from all cameras with a very general architecture that does not necessitate any strong geometric assumptions. Besides, when compared with CVT, we observe that LaRa obtains better results in the setting with finer resolution ($+10\%$ in Setting 1 vs. $+8\%$ in Setting 2).

Since our attention mechanism does not rely on all-to-all attention between camera images and BEV map as CVT does, LaRa can directly decode to the final BEV resolution which helps for fine prediction at a high resolution.

## 4.2 Model ablation and sensitivity to hyper-parameters

**Input and Output-level embeddings.** To assess the contribution of the geometric embeddings that we use, we compare the different choices at both the input and output level in Table 2. As hypothesized, embedding the geometric relationship between cameras in the input is better suited for our task than the generic sine and cosine spatial embeddings. The additional camera index, while performing better than Fourier feature alone, is not enough to link pixels across cameras. For the output query embedding, the combination of normalized coordinates and radial distance gives the best results. This simple choice outperforms both the Fourier features [25, 28] and learned embeddings [25, 27] that also have the disadvantage of increasing the number of parameters.

**Sensitivity to hyper-parameters.** To delve into the influence of hyper-parameters, we conduct a sensitivity analysis in Figure 2 where we vary the number $N$ of latent vectors, their dimension $M$ and the number of self-attention blocks $L$. We clearly observe that the performance increases with the number of latent vectors used. This is expected as it is the main parameter controlling the attentional bottleneck between input and output. Such a parametrization allows for an easy tuning of the performance/memory trade-off. We observe no clear correlation between the dimension $M$ of latent vectors, the number $L$ of self-attention layers, and the obtained IoU performance. This indicates that our architecture is not too sensitive to these hyper-parameters and can work efficiently with a wide range of values for these parameters. Although we obtain better results with 512 latent vectors, we use a maximum of 256 to stay in the same computational regime as the baseline we compare against; training with 512 latent vectors requires 32GB GPUs.

---

[2]https://github.com/valeoai/LaRa

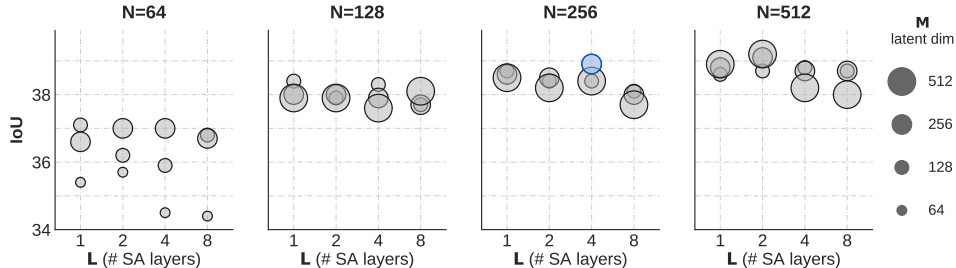

Figure 2: **Sensitivity study of LaRa to hyper-parameters.** We vary the number of latent vectors ($N$), their dimension ($M$), and the number of self-attention layers ($L$) and report IoU performances.

## 4.3 Study of attention

As quantitatively studied in Section 4.2, embedding camera rays impacts significantly the performance of LaRa. By analyzing the input-to-latent attention map, we further investigate the geometric reasoning of LaRa in Figure 3. In this figure, we show two representations of the attention: a re-projection of the attention in the camera-space (left) and a top-view projection of the attention in polar coordinates by collapsing, i.e., averaging the vertical dimension (right). In the latter, the radial distance is proportional to the attention level and shows the directions the network attends the most.

The study is conducted at three different levels. First, for a couple of one latent vector and one attention head ($n = 10$, $h = 5$ and $n = 50$, $h = 30$), among $N = 256$ possible latents and $H = 32$ possible attention heads. Second, for one latent vector and the averaged attention from all attention heads ($n = 10$, $h =$ avg and $n = 50$, $h =$ avg). Third, for one attention head and the averaged attention over all latents ($n =$ avg, $h = 5$ and $n =$ avg, $h = 30$). From these three settings, we note the followings: First, the attention map between one latent vector and one attention head targets a specific direction (about a 90° field of view). Additionally, it can be clearly observed that the attention is continuous across cameras, proving the network is able to retrieve the pixel relationships between views. Second, while one attention head fires in a specific direction, the attention averaged over all the heads for one latent vector spans over half of the scene. This allows one latent vector to extract long-range context between views with the capacity to disambiguate them. Third, the attention for one head aggregated over all the latent vectors covers all directions, suggesting that the latent vectors contain all of the directional information and that the whole scene is attended across the latents. To summarize, by integrating early multi-view geometric cues instantiated by camera rays embedding (Section 3.1), we show that LaRa learns to reason across views. We also provide quantitative evidence in the supplementary material.

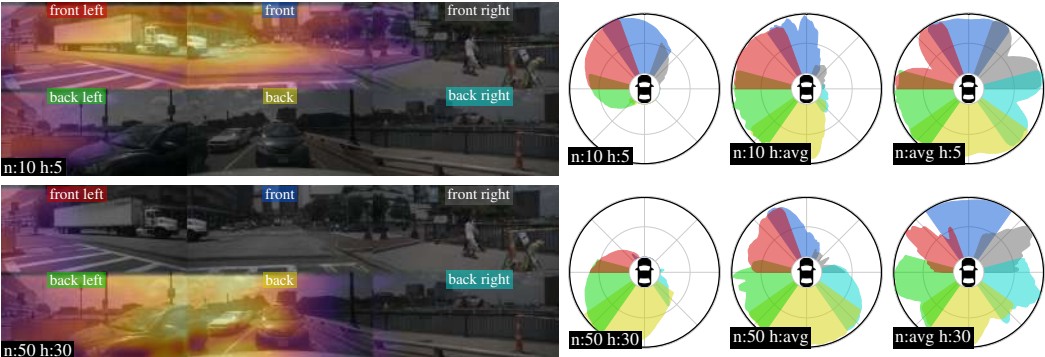

Figure 3: **Input-to-latent attention study.** Attention maps analysis for a network using 256 latents and 32 attention heads. The attention for one attention head and one latent is shown on the left superimposed with RGB images. The polar plots represent the directional attention intensity for one (or the average) attention head with one (or the average) latent vector. The radial distance is proportional to the attention level and shows the directions the network attends the most.

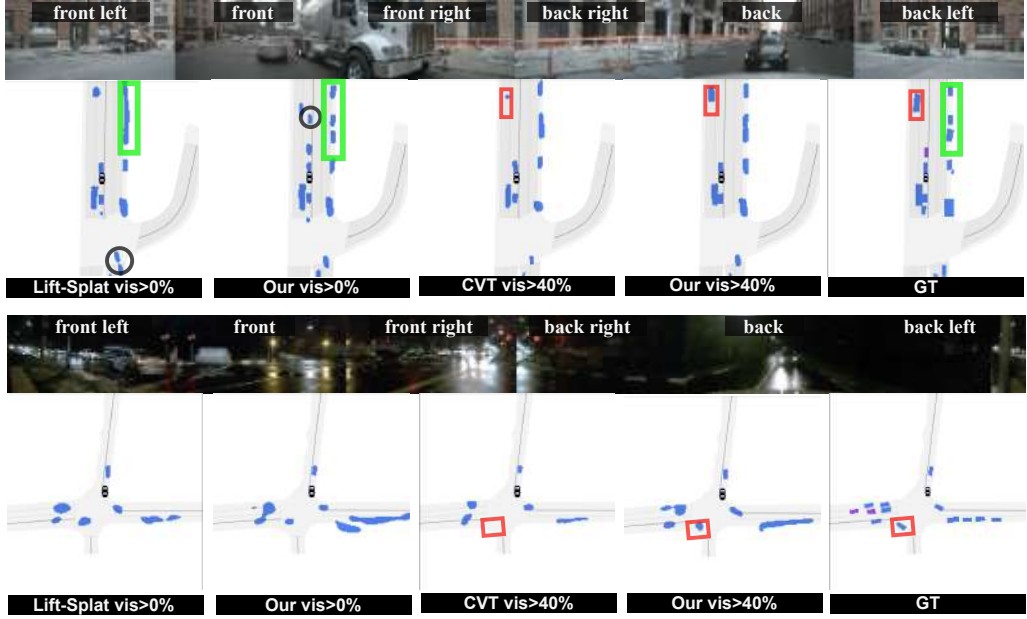

Figure 4: **Qualitative results on complex scenes.** We show the six camera views surrounding the vehicle along with segmentation map ground-truth for reference. In the ground-truth (GT) map, vehicles are shown in blue (visibility $> 40\%$) or purple (visisibility $< 40\%$). The ego vehicle is located in the center and facing downwards. We show results for our two baselines [4, 17]. For a fair comparison, we always compare using their respective level of visibility. Setting 2 is used.

## 4.4 Qualitative Results

We show the segmentation results of two complex scenes in Figure 4. For a fair comparison, we use our model trained with visiblity $> 40\%$ against CVT and $> 0\%$ against Lift-Splat. Compared to LaRa, CVT missed two objects, one at a long distance and the other in the dark (red box). We also estimate the boundaries of the vehicles better than Lift-Splat (green box). Interestingly, models trained on all vehicles (visibility $> 0\%$) tend to hallucinate cars in occluded or distant regions (highlighted with black circles in the figure).

## 5 Conclusion

We presented LaRa, which leverages transformer-based architectures and encoder-decoder models, with respectively efficient deep cross- and self-attentions as well as an explicit control on the computation and memory footprint thanks to decoupling the bulk of the processing from the input and output resolution. By incorporating ray embeddings into LaRa, we augment semantic features with geometric cues of the scene and show that this leads to multi-view stitching.

**Limitations.** Our model operates on camera inputs only. Thus, in adverse conditions, e.g., with glares and darkness, its performance remains limited. To better handle these challenging situations, one avenue of improvement would be the extension of LaRa to handle complementary modalities, e.g., coming from LiDARs or radars.

**Broader impacts.** LaRa demonstrates that the geometry and semantics of a complex scene can be compacted in a small collection of latent vectors. We believe that this formulation would allow for efficient temporal reasoning. Currently, the temporal modeling is done in the BEV space, which is high resolution and mostly represents empty space [5, 29].

**Acknowledgments**

This work was supported in part by the ANR grants AVENUE (ANR-18-CE23-0011), VISA DEEP (ANR-20-CHIA-0022), and MultiTrans (ANR-21-CE23-0032). It was granted access to the HPC resources of IDRIS under the allocation 2021-101766 made by GENCI.

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
