# OpenReview forum: "LaRa: Latents and Rays for Multi-Camera Bird’s-Eye-View Semantic Segmentation"
_robot-learning.org/CoRL/2022/Conference — CoRL 2022 Poster_

### Official Review · Reviewer_u5Wy · 2022-07-31

**Originality:** Fair
**Technical Quality:** Good
**Clarity Of Presentation:** Good
**Impact:** 3

**Recommendation:**

Weak Accept: I recommend accepting the paper, but will not argue for my recommendation if the majority of other reviewers have a different opinion.

**Summary:**

This paper proposes an efficient transformer-based encoder-decoder model for BEV vehicle semantic segmentation from camera-only inputs. It first encodes the semantic features from input images and camera intrinsic information into a set of latent variables with a system of cross-attention. These latent features are then processed with self-attention blocks and cross-attention with BEV queries to obtain the final semantic segmentation results. Experimental results demonstrate the efficacy and efficiency of the proposed method.

**Issues:**

See weaknesses.

**Quality Of The Limitations Section:**

Additional details required

**Reviewer Expertise:**

5: The reviewer is absolutely certain that the evaluation is correct and very familiar with the relevant literature

**Robotics Focus:**

Highly relevant to robotics but no hardware experiments

**Strengths And Weaknesses:**

Strengths:
- The paper is well-written and the basic idea is easy to follow.
- Encoding the input images into a set of latent vectors can avoid the significantly increasing computational burden caused by high-resolution PV and BEV features.
- Related works are comprehensively analyzed and compared.
- The qualitative visualization in Sec. 4.3 is interesting.
- Experiments show state-of-the-art results compared to previous works.

Weaknesses:
- This method is general and should be able to be applied to other tasks, such as map segmentation (not only vehicles) and 3D detection. There are more competitive entries and more baselines that can be compared in those benchmarks. More experimental results on those tracks can distinguish this paper from others better and make the conclusion more convincing.
- Questions about the encoding of camera calibration parameters:

a. There is another related work required to be compared and discussed: PETR: Position Embedding Transformation for Multi-View 3D Object Detection. This work also encodes similar information from the input perspective and achieves improvements.

b. How is x formulated in Eq. (1)? Except for the pixel coordinates, the depth should also be considered to make it capable of being transformed to 3D space? And why here disentangle the rotation and translation when constructing such embeddings in Eq. (1) and (2)? It should be more common to use both to directly calibrate all these points of different camera views into a unified 3D space? Here may need more explanations.

**Summary Of Recommendation:**

This paper proposes an effective framework for BEV vehicle semantic segmentation. It comprehensively analyzes previous works and improves several technical aspects over them. Experimental results can demonstrate its efficacy. Nevertheless, there are still questions in terms of the design details of camera calibration embeddings and more experiments on other relevant competitive benchmarks. I would vote for weak accept temporarily and would consider adjusting the score according to the authors' feedback on my questions.

---

> ### Author Response · Authors · 2022-08-18
> **First response to reviewer**
>
> Thank you for the helpful insights and remarks.
> Based on your suggestion, we are working on an extension to other tasks and a comparison against the embedding used in PETR.
> We will come back to you on these points within a week with concrete results and further discussions.
> In the meantime, we take the opportunity to clarify the other points on our geometric encoding below.
>
> **Q1. Geometric encoding: should the depth be considered ?**
>
> **A1.** The goal of our ray embedding is to encode the underlying 3D information of the pixel itself, and not the 3D information of the object seen by the pixel. The depth of the object observed at a pixel is implicitly estimated by the network.
> Precisely, pixels are 2D coordinates in the image plane and the camera's calibration tells where their associated individual sensors physically lie in the 3D world.
> The intrinsics define how a pixel relates to the camera center following a pinhole projective model.
> The extrinsics define the position and orientation of the camera center relative to the ego-car reference frame.
> By composition, the intrinsics and extrinsics fully define the 3D position and orientation of a pixel relative to the ego-car reference frame.
> This will be clarified in the paper.
>
> **Q2. Geometric encoding: clarification of the disentanglement**
>
> **A2.** Our use of the letters '$r$' and '$t$' may be the origin of a misunderstanding: $r$ is not a rotation per se. We encode *rays* (and not *points*) and they are fully defined by their origin ($t \in \mathbb{R}^3$) and direction ($r \in \mathbb{R}^3$). The vector $r$ defines the direction of the ray originating at the camera center ($t$) and passing through the pixel it encodes. Each pixel has a unique ray encoding, where $r$ helps to differentiate between pixels and define their viewing direction, while $t$ defines the camera to which the pixel belongs.
> This will be clarified in the paper.
>
> **Q3. Geometric encoding: similarity with PETR encoding**
>
> **A3.** In PETR, the embedding of each pixel is computed by sampling its ray given $D$ predefined depths. The 3D coordinates of the $D$ sampled points along the ray are normalized, concatenated, processed by an MLP and summed with the visual features. Conceptually, the embedding is a way to indicate to the network ``*this pixel can observe these 3D points in the camera frustum space*''.
>
> The embedding in PETR differs from as it is is limited by the sampling resolution (i.e., the $D$ predefined depths), as computation and memory footprint increase linearly with respect to $D$. In contrast, we showed that our constant-complexity embedding is effective as a 3D positional embedding.
>
> In addition to the quantitative comparison against the positional encoding used in PETR that we are currently investigating, we will include the above discussion in the paper.

---

> > ### Author Response · Authors · 2022-08-26
> > **Second response to reviewer**
> >
> > **Q4. Expanding to other tasks**
> >
> > **A4.** In this work we aimed to design a generic multi-camera BEV-mapping architecture. The semantic segmentation task (using simple decoders and no post-processing) allowed us to design, validate and ablate the different building blocks of LaRa in an intuitive manner. On top of LaRa, we are considering different research directions in view of a journal submission.
> >
> > Nevertheless, following the suggestion of reviewer T5RB, we tackled the driveable area segmentation task, also addressed by CVT [17]. Contrary to vehicle segmentation, this task requires the network to do "amodal completion" to a high degree, i.e., to correctly estimate regions of the road despite parts of it being severely occluded.
> >
> > We followed the protocol of CVT for this segmentation task; the ground truth is generated using HD-map's polygons from the dataset. We used the same hyperparameters in our framework set for the vehicle segmentation task, with a minor difference to the learning rate: we divide it by a factor 10 after 15 epochs (compared to a constant learning rate for vehicle segmentation).
> >
> > Results for this additional task comparison to CVT are given below, and qualitative examples are in the file attached. When compared with CVT, we observe that LaRa achieves better performance (+0.9).
> >
> > | Method  | IoU   |
> > |--|--|
> > | CVT      | 74.3  |
> > | LaRa (ours)  | **75.2** |
> >
> > Table 1: Intersection-over-Union (IoU) for driveable area segmentation on nuScenes.
> >
> > We will include this extension in the paper. Additionally, our code and the model's weights will be released upon publication to ease reproducibility.
> >
> > **Q5. Geometric encoding - comparison to PETR**
> >
> > **A5.** We now provide quantitative results to compare PETR embedding against our ray embedding. We trained our model with PETR input embedding in place of ours. The results below show that our ray embedding performs better. This will be included in the updated paper.
> >
> > | Embedding | IoU   |
> > |--|--|
> > | PETR    | 34.8  |
> > | Cam. rays (ours)  | **35.4** |
> >
> > Table 2: Intersection-over-Union (IoU) for vehicle segmentation on nuScenes.

---

### Official Review · Reviewer_Zycj · 2022-08-01

**Originality:** Very Good
**Technical Quality:** Excellent
**Clarity Of Presentation:** Excellent
**Impact:** 4

**Recommendation:**

Strong Accept: I recommend accepting the paper and will argue for my recommendation even if other reviewers hold a different opinion.

**Summary:**

This paper a method for vehicle segmentation in BEV space for use in autonomous driving. The system takes in images from multiple cameras, feeds them through a CNN to extract features, and then combines the features with ray embeddings (a representation of the ray that captured each pixel). The ray embeddings encode the spatial relationship between the cameras. The fused representation goes into a transformer, and outputs a BEV map which is further refined using a CNN. The authors show the method outperforms baselines, and also conduct several interesting ablation studies justifying various design choices.

**Issues:**

- In multiple places, need to change 32Go to 32Gb

**Quality Of The Limitations Section:**

Limitations are addressed clearly

**Reviewer Expertise:**

3: The reviewer is fairly confident that the evaluation is correct

**Robotics Focus:**

Highly relevant to robotics but no hardware experiments

**Strengths And Weaknesses:**

Strengths:
- The latent representation is less computationally expensive compared to methods that map densely between input image pixels and BEV pixels
- Does not require BEV projection, which can lead to compounding errors
- The model does not require a hardcoded BEV resolution or geometric assumptions
- Interesting analysis showing that the attention heads of the learned model cover many directions

Weaknesses:
- The performance of the system does not seem ready for use in such a safety critical application area


**Summary Of Recommendation:**

This paper proposes a novel method for BEV vehicle segmentation that outperforms other methods with a simpler and less computationally intensive approach. Also, the use of ray embeddings in this system seem likely to be of interest for other systems tackling this problem.

---

> ### Author Response · Authors · 2022-08-18
> **Response to Reviewer**
>
> Thank you for the review and positive feedback.
>
> **Q1. Readiness for safe deployment**
>
> **A1.** We agree that the current system is not ready for use in a level-5 autonomous driving stack that requires an almost perfect accuracy, reliability, and robustness in predictions.
> That being said, we believe that our model can already play a key role in robotic applications that assist users. For example, this includes the estimation of parking slot occupancy or triggering emergency alarms in case of unexpected traffic congestions.
> In the future, we will strive to make our system even more accurate, for example by integrating temporal information to reduce occlusion errors or by combining LiDAR/Radar data to make predictions more robust in case of adverse conditions.
>
> **Q2. Spelling corrections**
>
> **A2.** Thank you for pointing this out, we will make sure to correct this in the final manuscript.

---

### Official Review · Reviewer_T5RB · 2022-08-03

**Originality:** Good
**Technical Quality:** Very Good
**Clarity Of Presentation:** Good
**Impact:** 3

**Recommendation:**

Weak Accept: I recommend accepting the paper, but will not argue for my recommendation if the majority of other reviewers have a different opinion.

**Summary:**

This paper presents a method for computing Bird's Eye View (BEV) occupancy maps. Unlike methods that explicitly use geometric information or that compare all pairs of pixels, they use an encoder-decoder transformer model that contains a number of design decisions that allow it to be efficient and accurate. They fuse camera rays and what they term "semantic features" from a CNN. The rays incorporate geometric information by applying an MLP to the direction and camera center. Their model uses a fixed-size latent representation inspired by general-purpose architectures like the Perceiver. This avoids the quadratic penalty incurred by naive models. They evaluate on nuScenes vehicle segmentation, where they outperform several recent works. Through ablations, they find that the different aspects of the geometry and query embeddings improve results.

**Issues:**

If time permits, I would be interested in seeing how the model performs on other BEV prediction tasks.

**Quality Of The Limitations Section:**

Limitations are addressed clearly

**Reviewer Expertise:**

3: The reviewer is fairly confident that the evaluation is correct

**Robotics Focus:**

Highly relevant to robotics but no hardware experiments

**Strengths And Weaknesses:**

Strengths:
- Their approach is simple. The fixed-size latent is a nice way of addressing the quadratic complexity incurred by prior work.
- The ablation experiments validate the different components of their approach.
- The approach outperforms previous work such as CVT and Lift-splat.
- They seem to have made an attempt to control for differences in resolution and other experimental details (although I'm not an expert on this benchmark and other reviewers may be better-suited to evaluating this).

Weaknesses:
- The evaluation is a bit limited, covering only one task (vehicle segmentation). It would be helpful to expand this to other tasks considered in prior work (eg. the other comparisons in CVT [17]).
- I found the description of the latent representation (Section 3.2) to be sparse on details, although it is described more precisely in the supplemental material. It would be helpful to provide slightly more detail in the main text, given that this is one of the main sources of novelty.
- The approach is somewhat incremental. Most of the improvement comes from fairly well-known architectural changes.
- The conclusions in the attention study (Section 4.3) seem slightly overstated (eg. the claim "proving the network is able to retrieve the pixel relationships between views"). This study does not seem to have been conducted very systematically, and consists mostly of qualitative results.


**Summary Of Recommendation:**

Overall, I'm borderline on the paper but lean towards acceptance. While the approach is incremental, the architecture's design decisions seem reasonable, and this is an interesting application of a fixed-size latent space. The experiments suggest the model works well, though I will defer to other reviewers in assessing the soundness of the experimental design.

---

> ### Author Response · Authors · 2022-08-18
> **First response to reviewer**
>
> Thank you for the helpful insights and remarks. Based on your suggestion, we are considering to compare against CVT on the driveable area segmentation task, as well as a quantitative study of the attention continuity across cameras. We will come back to you on these points within a week with concrete results and further discussions. We now take the opportunity to clarify the other points below.
>
> **Q1. Additional information on the latent representation**
>
> **A1.** Thank you for pointing this out. In the updated version, we will clarify the inner workings of the latent representation in the main paper, building on elements from the Supplementary.
>
> In particular, we will clarify that our architecture integrates three attention modules: (i) the cross-attention between latent vectors and input features (Sec. 3.2); (ii) a sequence of self-attention on the latent vectors (Sec. 3.2); (iii) a cross-attention between BEV query and latent vectors (that will remain in Sec. 3.3). Besides, we will make more refined references to Figure 1 to guide the reader throughout Sec. 3.2.
>
> **Q2. Contributions**
>
> **A2.** Here, we would like to emphasize the different contributions we make in our work.
>
> 1. We propose the first method that can encode images from multiple cameras mounted on a robot/vehicle into a compact latent space. Not only does it enable a precise control of the model's memory and computation footprint, but also we show that this latent space holds rich semantic and spatial information about the scene and that it can be exploited and reprojected into spaces of very different nature (Camera to BEV in our case). We believe this kind of general system can be of interest for the community.
>
> 2. Moreover, we propose a new ray embedding for transformer-based architectures that are tailored for our task as we show that it helps to fuse multiple views taken from the different cameras. Besides, we present a new way to visualize cross- and self-attentions for multi-camera transformer-based systems.
>
> 3. Our design choices are validated with thorough experiments and ablations, and we show that the final system is not very sensitive to hyper-parameters, which holds the promise of making our approach easily applicable to other datasets and related tasks.

---

> > ### Author Response · Authors · 2022-08-26
> > **Second response to reviewer**
> >
> >
> > **Q3. Extension to other tasks (e.g., CVT [17])**
> >
> > **A3.** We worked on the driveable area segmentation problem as another task. Contrary to vehicle segmentation, this task requires the network to do "amodal completion" to a high degree, i.e., to correctly estimate regions of the road despite parts of it being severely occluded.
> >
> > We followed the protocol of CVT for this segmentation task; the ground truth is generated using HD-map's polygons from the dataset. We used the same hyperparameters in our framework set for the vehicle segmentation task, with a minor difference to the learning rate: we divide it by a factor 10 after 15 epochs (compared to a constant learning rate for vehicle segmentation).
> >
> > Results for this additional task comparison to CVT are given below, and qualitative examples are in the file attached. When compared with CVT, we observe that LaRa achieves better performance (+0.9).
> >
> > | Method | IoU   |
> > |---|---|
> > | CVT     | 74.3 |
> > | LaRa (ours) | **75.2** |
> >
> > Table 1: Intersection-over-Union (IoU) for driveable area segmentation on nuScenes
> >
> > We will include this extension in the paper. Additionally, our code and the model's weights will be released upon publication to ease reproducibility.
> >
> >
> >
> > **Q4. A quantitative study of attention consistency across cameras**
> >
> > **A4.** We propose a quantitative analysis to support our claim that "our network is able to retrieve the pixel relationships between views thanks to our ray embedding" (Sec. 4.3).
> >
> > To this end, we introduce a metric that directly quantifies the consistency and alignment of attention values across camera by analyzing behavior in "overlapping" regions, i.e., regions seen by two different cameras.
> > In the file attached we provide a visual description of this metric and its computation.
> >
> > In short, knowing the orientation of each camera, we compute the Mean Squared Error (MSE) of the directional attention intensity (Fig. 3) between cameras on their overlapping regions. This score is averaged for all the overlapping regions, latents and attention heads, and examples in the validation set. A score of zero indicates a perfect match of the attention levels on overlapping regions (i.e., across cameras). Results with this metric, reported below, show that our "Cam. rays" embedding is 10 times more 'consistent' across cameras than the baseline "Fourier + Cam. idx". This additional analysis will be included in the paper.
> >
> > | Embedding | MSE on overlap|
> > |---|---|
> > | Fourier + Cam. idx | 0.0896 |
> > | Cam. rays (ours) | **0.0068** |
> >
> > Table 2: Mean-Squared-Error(MSE) of the attention intensity on overlapping regions between cameras.
> >
> > Additionally, in the attached pdf, we provide qualitative examples of the "Fourier + Cam. idx" embedding to compare against our ray embedding. Contrary to the attention yield by our ray embedding, the one derived from the "Fourier + Cam. idx" embedding is much more spread out and less consistent across cameras.

---

### Meta-Review · Area_Chair_2V7W · 2022-08-15

**Recommendation:** Accept (Poster)
**Confidence:** 4

**Metareview:**

This paper presents an encoder-decoder transformer model for BEV vehicle semantic segmentation with camera-only inputs.

The technical quality is very good, and clarity and originality are good.

The strengths are:
- Well-written and ideas easy to follow,
- The proposed approach is quite efficient in addressing common computational issues,
- The ablation study validates the different components,
- There is sufficient comparison with SOTA.



**Best Paper Nomination:**

No

---

> ### Author Response · Authors · 2022-08-26
> **General response**
>
> We thank all the reviewers and the meta-reviewer for their insightful feedback and helpful suggestions.
> The clarifications and discussions we provide for each remark/question include the following:
>
> * We extended our architecture LaRa to segment the driveable area in bird's-eye-view and compared against CVT [17] on this task. We follow authors' setting and produce better results with no significant changes to our hyperparameters.
> * We consolidated the claim that "our network is able to retrieve the pixel relationships between view thanks to our ray embedding" by (i) providing a quantitative analysis of the attention continuity across cameras, and (ii) comparing quantitative and qualitative results between our "Cam. rays" embedding and the "Fourier + Cam. idx" baseline.
> * We detailed key differences between the PETR input embedding and our ray embedding. Additionally, we trained our model LaRa with the PETR input embedding and show that this variant is less effective than using our embedding.
>
> These discussions, additional analyses and new experiments will be included in the paper.